# Lipophilicity Determination of Antifungal Isoxazolo[3,4-*b*]pyridin-3(1*H*)-ones and Their N1-Substituted Derivatives with Chromatographic and Computational Methods

**DOI:** 10.3390/molecules24234311

**Published:** 2019-11-26

**Authors:** Krzesimir Ciura, Joanna Fedorowicz, Filip Andrić, Petar Žuvela, Katarzyna Ewa Greber, Paweł Baranowski, Piotr Kawczak, Joanna Nowakowska, Tomasz Bączek, Jarosław Sączewski

**Affiliations:** 1Department of Physical Chemistry, Faculty of Pharmacy, Medical University of Gdańsk, Al. Gen. J. Hallera 107, 80-416 Gdańsk, Poland; katarzyna.greber@gumed.edu.pl (K.E.G.); pawel.b1995@gmail.com (P.B.); jonowak@gumed.edu.pl (J.N.); 2Department of Chemical Technology of Drugs, Faculty of Pharmacy, Medical University of Gdańsk, Al. Gen. J. Hallera 107, 80-416 Gdańsk, Poland; joanna.fedorowicz@gumed.edu.pl; 3Department of Analytical Chemistry, University of Belgrade—Faculty of Chemistry, Studentski trg 12–16, 11000 Belgrade, Serbia; andric@chem.bg.ac.rs; 4Department of Pharmaceutical Chemistry, Faculty of Pharmacy, Medical University of Gdansk, Al. Gen. J. Hallera 107, 80-416 Gdańsk, Poland; petar.zuvela@nus.edu.sg (P.Ž.); p99p@gumed.edu.pl (P.K.); tbaczek@gumed.edu.pl (T.B.); 5Department of Organic Chemistry, Faculty of Pharmacy, Medical University of Gdańsk, Al. Gen. J. Hallera 107, 80-416 Gdańsk, Poland; jaroslaw.saczewski@gumed.edu.pl

**Keywords:** lipophilicity, isoxazolo[3,4-*b*]pyridin-3(1*H*)-one, isoxazolone, reversed-phase liquid chromatography, sum of ranking differences, QSRR analysis

## Abstract

The lipophilicity of a molecule is a well-recognized as a crucial physicochemical factor that conditions the biological activity of a drug candidate. This study was aimed to evaluate the lipophilicity of isoxazolo[3,4-*b*]pyridine-3(1*H*)-ones and their N1-substituted derivatives, which demonstrated pronounced antifungal activities. Several methods, including reversed-phase thin layer chromatography (RP-TLC), reversed phase high-performance liquid chromatography (RP-HPLC), and micellar electrokinetic chromatography (MEKC), were employed. Furthermore, the calculated log*P* values were estimated using various freely and commercially available software packages and online platforms, as well as density functional theory computations (DFT). Similarities and dissimilarities between the determined lipophilicity indices were assessed using several chemometric approaches. Principal component analysis (PCA) indicated that other features beside lipophilicity affect antifungal activities of the investigated derivatives. Quantitative-structure-retention-relationship (QSRR) analysis by means of genetic algorithm—partial least squares (GA-PLS)—was implemented to rationalize the link between the physicochemical descriptors and lipophilicity. Among the studied compounds, structure **16** should be considered as the best starting structure for further studies, since it demonstrated the lowest lipophilic character within the series while retaining biological activity. Sum of ranking differences (SRD) analysis indicated that the chromatographic approach, regardless of the technique employed, should be considered as the best approach for lipophilicity assessment of isoxazolones.

## 1. Introduction

Lipophilicity is one of the essential factors that determine the biological activity of drug candidates. It determines not only the transport of molecules through biological membranes but also their ability to undergo complexation with blood proteins and binding to receptors [1]. Knowledge of lipophilicity helps us understanding pharmacokinetic properties, including absorption, distribution, metabolism, and excretion (ADME) processes, as well as toxicity [2,3].

Although lipophilicity has been used since the 1960s as one of the main parameters in medical chemistry, the standardization of the high-throughput and reliable analytical procedure for its assessment is still required [4]. According to the International Union of Pure and Applied Chemistry (IUPAC), the operational definition of lipophilicity is the affinity of a molecule or a moiety for a lipophilic environment. It is commonly measured by its distribution behavior in a biphasic system, either a liquid–liquid or solid–liquid system [5]. 

The direct, “shake flask” method proposed by Hansch and co-workers employing *n*-octanol–water partitioning is laborious, time-consuming, and requires large amounts of organic solvent and absolutely pure substances. For this reasons, methods based on the solid–liquid partitioning such as reserved phase liquid chromatography (RP-LC) and electrometric techniques including micellar electrokinetic chromatography (MEKC) and microemulsion electrokinetic chromatography (MEEKC) are currently gaining popularity [6,7,8,9,10]. Simultaneously, in silico approaches dedicated to the calculation of partition coefficient (log*P*) are being intensively developed. 

The computational methods that provide quick information regarding lipophilicity are commonly utilized for the screening of chemical libraries. Hence, in silico approaches have considerable advantages when compared to experimental methods. They are significantly faster and cheaper, since their application requires neither laboratory experiments nor specialized equipment and chemical reagents. Another advantage of the calculation approach is a possibility to estimate lipophilicity during the design of drug candidates prior to their synthesis. There are several programs dedicated to log*P* calculation. Most of them are freely available online and generate results nearly instantaneously. 

In addition, recently presented reports prove the usefulness of ab initio methods and density functional theory (DFT) for lipophilicity assessments [11,12]. The latter approach is based on the calculated Gibbs energy differences for solvated molecules with respect to various phases, i.e., n-octanol and water.

Various substituted isoxazolone derivatives have demonstrated a broad spectrum of biological activities including antibacterial [13,14], antitubercular [15], fungicidal [16], antileukemic [14], anticancer [17], antioxidant [18], antiandrogenic activities [19], and many others [20]. 

The investigated isoxazolo[3,4-*b*]pyridine-3(1*H*)-ones derivatives (Figure 1) exhibit moderate antibacterial properties and pronounced antifungal activity (MIC < 6.2 µg/mL for *Candida parapsilosis*) [15,21,22]. The matter of new antifungal drugs has become a vital topic, due to the development of resistance to currently used antifungal drugs, as well as their high toxicity, thus rendering the treatment of fungal infections a major challenge for modern medicine.

This work pertains to an extensive study aimed at the assessment of the physicochemical properties of pyrido-isoxazolone derivatives with proven antifungal activity. For lipophilicity evaluations, several methods—including two modes of RP-LC, reversed phase thin layer chromatography (RP-TLC), and reversed phase high performance liquid chromatography (RP-HPLC)—were used. Furthermore, for lipophilicity assessment, MEKC was also employed. The calculated log*P* values were established with various freely and commercially available software packages and online platforms that include theoretical models based on pure atomic and/or fragment contribution approaches, properties dependent methods, and DFT computations. Chemometric methodology was applied in order to select the optimal tools for lipophilicity assessment of the studied isoxazolone derivatives. Finally, quantitative structure retention relationship (QSRR) analysis was performed in other to rationalize the link between the physicochemical properties of target compounds and their lipophilicity.

## 2. Results and Discussion

Taking into account the importance of lipophilicity from a medicinal chemistry perspective, the present study focuses on comparing the performance of methods for lipophilicity determination with regard to the aforementioned group of drugs candidates. 

### 2.1. Lipophilicity Estimation by Computational Methods

Table 1 lists all the computational log*P* parameters attained for the tested compounds using particular software programs. Chemical names of the target compounds are given in Appendix A, whereas 2D structures are presented in Figure 1. 

Significant differences in terms of minimum, maximum, and mean values of log*P*, depending on the theoretical approach applied, were found (Appendix A). Furthermore, it was noted that not all of the obtained lipophilicity parameters correlate with each other, as indicated in the presented correlation matrix (supplementary data excel file in the worksheet correlation matrix). 

These differences can be explained by the diversified nature of the algorithms employed in the utilized software programs. Generally, the following three main groups of theoretical methodologies can be distinguished: the atomic approach, the fragment contribution technique, and properties dependent methods. Classification of the investigated software based on algorithm type is presented in Table 2. 

The lowest mean value of log*P* among the investigated software platforms was obtained using Villar algorithms implemented into Spartan (Log*P*V Spartan). This method employs semi-empirical wave functions and a calculated overlap matrix, which includes type and number of lone pairs, along with the surface area of selected atoms (H, C, N, O, F, S, and Cl). Surprisingly, according to this program the maximum log*P* value of 4.72 was obtained for compound **12** (pyrido-isoxazolone substituted with the methylsulfonyl group). This result is in contradiction with the log*P* value of 0.69 calculated for the same structure with the alternative Ghose–Crippen algorithm available in Spartan (Log*P*C Spartan). Hence, this atomic approach–based method, which is parameterized for 110 atom types and includes correction factors, proved the compound described above to be the most hydrophilic in the series. Another striking example is compound **5**, for which the log*P* parameter obtained with fragment contribution–based software is 3.51 (Mlog*P*), whereas the lipophilicity index calculated by hybrid KOWWIN algorithm, implementing both an atom-based approach and fragment contribution framework, is significantly lower (1.88). 

The presented results indicate that the computational methods can be very unreliable with respect to certain functional groups. In consequence, the same chemical structure can feature completely different log*P* parameters depending on the algorithm applied.

Significant differences between log*P* values calculated for particular chemical structures can be considered as one of the major drawbacks of the computational methods. The above observation justifies the fact that the experimental methods are still preferred over the computational approaches and provides evidences of the importance of proper method selection in lipophilicity assessment processes. 

Alternative ab initio methods, based on DFT wave functions and solvation models can be utilized for calculation of the free solvation energy change for the analyte transfer between n-octanol and water phases. This approach is considerably less popular than the other theoretical techniques, and hence only a few reports proved usefulness of DFT calculations for partition coefficients estimation. Those publications concern lipophilicity assessments for various groups of chemical species, such as alcohols [12], ruthenium(II)-arene complexes [23], and organophosphate-type pesticides [11]. Table 3 summarized the calculated log*P* parameters using investigated DFT methods. 

Among the tested functionals, a good overall agreement between the DFT calculated log*P*s and chromatographically determined lipophilicity indexes (log*k*_w_) was found for PBE0 6-311G++(2df,2dp) (*R* = 0.680, *p* = 0.005). The outliers detected by 2.5 sigma rule are compounds **11**, **9**, and **12**. After exclusion of outliers from the obtained model, the value of the correlation coefficient (*R*) increases considerably (*R* = 0.895, *p* = 0.005). These results suggest that regular development of ab initio methods may lead to a substantial improvement of the aforesaid lipophilicity assessment procedure. The major problem in this approach is the selection of a suitable functional for calculation of molecules featuring particular chemical or functional groups. Undoubtedly, better understanding of this subject—as well as the application of self-learning algorithms, such as artificial neural networks, for selection of an appropriate functional—may contribute to dissemination and improvement of the ab initio lipophilicity assessment approach. 

### 2.2. Estimation of Lipophilicity by Chromatographic Methods

In parallel to computational methodologies, indirect approaches (mostly chromatographic) have been utilized for lipophilicity assessments [9,10]. This is attributed to the fact that lipophilicity governs the retention of molecules in RP-LC, which is one of the most popular modes of LC separation in the field of pharmaceutical chemistry [24].

The pyrido-isoxazolone derivatives were analyzed by RP-LC and micellar electrokinetic chromatography (MEKC). The experimentally determined chromatographic lipophilicity indices are summarized in Table 4.

The present study comprehends two types of reversed phase liquid chromatography setups—that is, RP-TLC and RP-HPLC techniques—were utilized. RP-TLC analyses were performed on two main reversed stationary phases, i.e., C_8_ and C_18_ modified silica. Mixtures of methanol and water were used as mobile phases according to Komsta’s recommendation [25]. However, in order to attain reasonable retentions for the investigated compounds on C_18_ bonded silica gel, higher amounts of methanol in the mobile phase were required when compared to the experiments performed with application of C_8_ modified silica. Generally, regular chromatographic behavior of the target structures was observed. The retardation factor (R_F_) increased systematically with increasing fraction of methanol in the mobile phase. As might be expected, the behavior of the analytes on both types of plates well reflects the Snayder–Soczewiński equation. These findings were confirmed by high values of correlation coefficient (*R*), determination coefficient (*R*^2^), significant F-Snedecor’s test (*F*) and small standard estimation error (*s*). The statistical figures for the obtained linearity of Snyder–Soczewiński equation are summarized in Appendix A. 

Subsequently, the intercorrelation between *m* and *R*_M_^0^ was verified in order to evaluate the hypothesis, that the analytes can be regarded from chromatographic point of view as a group of structurally similar compounds. The established linear correlations between slope and intercept of Soczewiński–Wachtmeister’s equation are presented below.

*R_M_*^0^ C_8_ = −0.925 (±0.058)*m* − 0.392 (±0.183)(1)

*R*^2^ = 0.912; *F* = 247.90 *p* < 0.0001; *s* = 0.174; *n* = 26

*R_M_*^0^ C_18_ = −1087 (±0.049)*m* + 0.694 (±0.149)(2)

*R*^2^ = 0.954; *F* = 497.62; *p* < 0.0001; *s* = 0.135; *n* = 26

The results obtained clearly indicate that the investigated isoxazolone derivatives constitute a series of chromatographically related congeners.

Routinely, lipophilicity measurements with RP-TLC are expressed as a *R*_M_^0^ values. However, other RP-TLC lipophilicity indexes, such as the mean value of *R*_M_, *C*_0_, and *PC*_1_, can be calculated on the basis of the retention data. Parameter m characterizes hydrophobic character of the studied compounds. All experimentally determined RP-TLC lipophilicity and hydrophobicity constants are listed in Table 4. The correlation matrix of all assessed parameters for the two investigated stationary phases are presented in the supplemented data sheet file. In general, all the obtained lipophilicity parameters are highly correlated, which indicates that they present similar information. Only the hydrophobicity parameter m shows significantly lower correlations with mean value of *R*_M_, *C*_0_, and *PC*_1_. Furthermore, correlation between RP-TLC lipophilicity indices and HPLC determinant log*k*_w_ was found. This result suggests that, for this class of chemical compound, RP-TLC can be used as pilot method for optimization of their RP-HPLC separations. 

Surprisingly, RP-LC chromatographic indexes and log*k*_MEKC_ were uncorrelated with each other. The correlation between RP-LC chromatographic indexes and log*k*_MEKC_ was expected due to theoretical assumptions that in both techniques the retention depends on lipohilicity [26]. Although, the MEKC background electrolyte (BGE) contained TRIS/HEPES buffer at pH 7.4, the investigated structures should not be subjected to ionization, except for compounds **1**, **14**, and **23**. As the pK_a_ value of compound **14** is 6.9 [15], in alkaline pH, the isoxazolones are subjected to acid dissociation and, hence, anion species are formed. Nonetheless, due to the presence of two ambident nucleophilic nitrogen atoms (N1 and N7 for **14** or N1 and N9 for **1** and **23**), these compounds can exist as mixtures of prototropic 1H-oxo and 7H/9H-oxo tautomers. However, the DFT calculations reveal that, in a solution of high relative permittivity (e.g., water, DMF), the 7H/9H-oxo forms are thermodynamically favored over the 1H-oxo tautomers [15,21].

The differences between the retention of the studied isoxazolone derivatives in MEKC and RP-LC experiments can be explained by the chemical character of the sodium dodecyl sulfate (SDS) micelles. SDS is an anionic surfactant that forms negatively charged micelles. In the case of the investigated structures, other lipophilic–hydrophobic interactions between solutes and micelles can occur. This phenomenon has been thoroughly studied and described in the section dedicated to QSRR.

Generally, the chromatographic lipophilicity parameters are roughly correlated with the calculated log*P*s obtained using both DFT calculations and software (correlation matrix in the supplementary datasheet excel file). The highest correlation coefficients (*R*) were observed for *R*_M_^0^ and *m* C_8_ plates and Clog*P* (Chem Draw). Concurrently, the lack of correlations between log*k*_MECK_ and the calculated log*P*s should be emphasized.

Analysis of the presented results indicates that the most lipophilic compound within the studied series is compound **2**, i.e., N1-benzyl substituted quinolino-isoxazolone. Hence, this structure exhibited the highest log*k*_w_ and *R*_M_^0^ in HPLC and C_18_ TLC measurements. Likewise, compounds **6**, **11**, **13**, and **18** bearing large alkyl and benzyl substituents displayed pronounced lipophilic properties. The least lipophilic character demonstrated the derivative **14**, i.e., pyridino-isoxazolone without any substituent on the ring nitrogen atom. The correlation between lipophilic properties and antifungal activities of the studied compounds was analyzed. The previously reported MIC data for five different Candida species (*C. albicans* ATCC 10231, *C. glabrata* ATCC 66032, *C. lusitaniae* ATCC 34499, *C. parapsilosis* ATCC 22019, *C. tropicalis* ATCC 750), assessed for all compounds except structure **13** [15,21,22], are presented in Appendix A. The lack of correlation between any of the lipophilicity indexes, both calculated and chromatographically determined, and antifungal activities should be emphasized. The most active compounds **14**, **15**, and **16**, although of diversified lipophilic attributes, log*k*_w_ in the range from 2.81 to 3.70, show similar antifungal properties. These results indicate that other of lipophilicity molecular properties affect antifungal activities of the studied isoxazolo[3,4-*b*]pyridine-3(1*H*)-ones and suggest a non-specific mode of antifungal action.

### 2.3. Multivariate Analysis of Antifungal Activity, Chromatographic Lipophilicity Indices, and Computationally Estimated logP Values

In order to investigate differences between chromatographic and computational lipophilicity measures and antifungal activities of the studied compounds, principal component analysis (PCA) was performed. PCA is one of the basic multivariate techniques which provides an insight into data structure, similarities and dissimilarities of variables, disposition of objects, tendencies for their grouping, and outlying effects. Therefore, it is usually carried out during the data exploration step. PCA transforms a huge number of variables into a significantly smaller set of new orthogonal variables called principal components (PC). The results of PCA are usually illustrated as a pattern of objects (score plot) or variables (as points in two-dimensional plots). Figure 2 presents the loading diagram of PC_1_ versus PC_2_, where investigated lipophilicity indexes both chromatographically determined and calculated together with antifungal activity are projected. Compound **13** was excluded from this analysis because microbiological data for this derivative were not available.

The first two PCs included 65.46% of the overall data variability. The greatest differences in the value of the PC_1_ can be observed between the RP-TLC parameter m, and PC_1_ for both stationary phases and other investigated parameters. Other RP-TLC parameters as well as log*k_w_* determined by HPLC are located together with computationally estimated log*P* values. However, the value of PC_2_ set apart log*P*s calculated by means of DFT algorithms. Only the experimentally obtained log*k*_MEKC_ is located centrally, distant from other lipophilicity parameters. Furthermore, all microbiological data, expressed as 1/log MIC, are grouped closely together, albeit at a considerable distance to the other lipophilicity parameters, both experimental and computational.

### 2.4. Comparison of Computationally and Chromatographically Derived Lipophilicity Indices by the Sum of Ranking Differences (SRD)

Computational and chromatographic methods have been analyzed by the sum of ranking differences in order to rank, group, compare, and select the best lipophilicity measures. Although some exploratory, unsupervised chemometric tools such as PCA and cluster analysis may reveal similarities among lipophilicity indices [27], they do not provide the possibility for selection of the best and the worst methods. Also, PCA deals only with a fraction of data variability, while the SRD takes the entire information pool. The superiority of the SRD method in comparison of lipophilicity measures has been demonstrated on several occasions [28,29,30].

In the case of the SRD-CRRN ranking of standardized lipophilicity measures (Figure 3a), the methods with smallest SRD values, i.e., the closest to the consensus, are Xlog*P*3, *m*, and *R*_M_^0^ obtained on C_8_-modified silica. These are closely followed by the rest of chromatographic and computational estimations, most of which are located on the left side of the plot, far from the random distribution curve, which makes them statistically significantly ranked. However, the retentions obtained under MEKC conditions are the worst measures grouped with the log*P* values estimated by ChemDraw. They fall under the random distribution curve and are unable to rank the studied compounds according to their lipophilic character better than a chance. The similar ranking was observed in the case of interval scaled and rank transformed data.

Sevenfold cross-validation followed by ascending ordering of the SRD medians and non-parametric pairwise comparisons of methods (by the sign test and matched-pairs test) reveals two large groups of lipophilicity indices separated at the predefined significance level of *p* = 0.05 (Figure 3b). In the first, much smaller group, only three chromatographic indices and most of the common computational methods can be found. In the second group, the rest of chromatographic indices and all of the DFT computational methods with remaining computational approaches are segregated. Only four indices—Vlog*P*, AClog*P*, log*P* Chem Draw, and log*k*_MEKC_—are located at the very end, and are all mutually separated at the predefined significance level of *p* = 0.05 and from the rest of the methods.

Since the sevenfold cross-validation introduces variability in the SRD values it is possible to decompose such variability to factors that can affect SRD ranking, and test their significance by ANOVA. In this particular case 651 SRD values were collected (31 lipophilicity measures ×3 data pretreatment methods ×7 repetitions) and subjected to ANOVA.

The full interaction model without quadratic terms (Equation (3)) was defined for two factors of particular interest.

(3)$$SRD=b_{0}+b_{1}F_{1}+b_{2}F_{2}+b_{12}F_{1}F_{2}$$

*F*_1_ represents data transformation (three levels corresponding to STD, IS, and RNK), and F_2_ describes different lipophilicity descriptor types (nine levels corresponding to computational methods based on fragment contributions (Comp_F), atomic contributions (Comp_A), property-based ones (Comp_P), mixed (Comp_M), and those based on DFT calculations (Comp_DFT), chromatographic indices (TLC_C_8_, TLC_C_18_, HPLC_C_18_, and MEKC)).

Statistical significance of factors is summarized in Table 5. Only types of lipophilicity descriptors significantly affect the outcome of SRD scores. As expected, data treatment, as well as the cross-coupling term are statistically insignificant at *p* = 0.05.

Plotting the means of factor levels and 95% confidence intervals (Figure 4) reveals that the SRD scores are the lowest in the case of chromatographic descriptors obtained on C_8_- and C_18_-modified silica, regardless to chromatographic technique employed (TLC or HPLC). Considering the fact that the lower the SRD scores are, the better are the methods, these chromatographic methods should be considered as the best ways for lipophilicity estimation. MEKC gives the highest SRD values, i.e., it is the worst method for lipophilicity determination. All computational methods, including those based on DFT, are of the same performance. The Fisher’s post hoc test can differentiate between all these three groups at the predefined significance of *p* = 0.05. However, Tukey’s Honest significant difference methods can only separate MEKC from the rest.

### 2.5. Quantitative Structure Retention Relationships Analysis of Chromatographically Derived Lipophilicity Indices

In pursuance of prediction of the retention factors and cognition of the retention mechanism for the four techniques used to estimate lipophilicity indices—RP-TLC (both C_8_ C_18_ plates), HPLC, and MEKC—QSRR analysis was employed. First, the application of the stringent criteria detailed in the experimental section resulted in a reduction of the initial matrix of 2848 descriptors to 44, 43, 43, and 38, for TLC C_8_
*R*_M_^0^, C_18_
*R*_M_^0^, HPLC log*k**_w_*, and log*k*_MEKC_ QSRR models, respectively. After 1000 iterations of GA-PLS, the first three consensus models comprised of 17, whereas the log*k*_MEKC_ QSRR model comprised of 12 molecular descriptors. Figure 5A–D depicts the percentage of selected molecular descriptors (%Selection) upon 1000 iterations of GA-PLS for each modeled end point.

The number of PLS latent variables (LVs) was comprehensively optimized. Optimal number of LVs for the first three GA-PLS QSRR models was four (RMSECV of 0.311, 0.353, and 0.197), whereas for the log*k*_MEKC_ model, the optimal number of LVs was five with an RMSECV of 0.104 (Appendix A). All the models were found to be strongly statistically significant (using CV-ANOVA) with *p* values < 0.0001 (Appendix A) and exhibited strong predictive ability as evident from Figure 6A–D with mean RMSE values (across the training and validations sets) of 0.126, 0.128, 0.183, and 0.081, TLC C_8_
*R*_M_^0^, C_18_
*R*_M_^0^, HPLC log*k*_w_, and log*k*_MEKC_ QSRR models, respectively. On the other hand, mean RMSE values of the full PLS model were 0.131 (6 LVs, RMSECV of 0.660), 0.132 (6 LVs, RMSECV of 0.423), 0.156 (5 LVs, RMSECV of 0.392), and 0.077 (6 LVs, RMSECV of 0.200), TLC C_8_
*R*_M_^0^, C_18_
*R*_M_^0^, HPLC log*k*_w_, and log*k*_MEKC_ QSRR models, respectively. The full PLS model exhibited markedly higher errors, especially of LOO-CV, except for mean RMSE of the QSRR models for HPLC log*k**_w_* and MKEC log*k*. The LOO-CV predictive performance (on the training set) is depicted in Appendix A. It can be observed that the trend of LOO-CV (training set) predictive performance for all the models except the consensus MEKC GA-PLS model (Appendix A) matches the predictive performance trends shown in Figure 5. The inconsistency between the training RMSE and LOO-CV RMSECV (for the training set) for the MEKC GA-PLS points to a potential over-estimation of the training error using LOO-CV. These results indicate that the established models can be successfully used for prediction on chromatographic indexes of pyrido-isoxazolone derivatives based on the computational descriptors. Nevertheless, the MEKC GA-PLS model should be applied with care.

Another benefit of QSRR analysis is the possibility of getting insight into the molecular mechanism of retention. For this, the molecular descriptors that affect chromatographic lipophilicity indexes have been determined. Appendix A summarizes full names and classes of descriptors used to build GA-PLS QSRR models.

In the case of RP-LC, the importance of two groups of molecular descriptors—geometry, topology, and atom-weights assembly (GETAWAY) descriptors and weighted holistic invariant molecular (WHIM) descriptors—should be highlighted. GETAWAY descriptors provide information regarding 3D-molecular geometry afforded by the molecular influence matrix (MIM) and atom relatedness by molecular topology, with chemical information obtained applying various atomic weightings (atomic mass, polarizability, van der Waals volume, and electronegativity, together with unit weights) [31]. This class of descriptors implement MIM, which is the matrix representation of molecules denoted by hydrogen atoms and constituted by the centered Cartesian coordinates x, y, z [32]. WHIM descriptors also belong to the group of 3D-molecular descriptors. This class of molecular descriptors contain information with regard to the whole 3D structure in terms of size, shape, symmetry, and atom distribution [33]. The calculation algorithms include realization of PCA based on the centered molecular coordinates. Although these two groups of descriptors are similar to each other, the WHIM descriptors reflect the holistic representation, whereas GETAWAY descriptors more effectively present information with respect to portions of the molecular structures [34]. It should also be emphasized that there were significant contributions to retention in RP-LC to descriptors that are weightings of polarizability, such as E2p, GATS5p, R3p, R4p, and R7p+.

The obtained GA-PLS QSRR models also indicate differences between RP-LC and MEKC retentions of studied isoxazolone derivatives. In case of MEKC, the crucial descriptors that affect the retention are HATS1e and CATS3D_08_AA. HATS1e that belong to GETAWAY descriptors class weighted by Sanderson electronegativity. Hence, these descriptors can explain differences between retention of studied compounds in RP-LC and MEKC. For example, HATS1e variability may influence interactions between negatively charged SDS micelles and electronegative groups of solutes. Another descriptor that significantly affects retention under MEKC conditions belongs to CATS3D descriptor class. The chemically advanced template search (CATS) descriptors are introduced as a pharmacophore/biophore model based on the cross-correlation of generalized atom types [35]. The CATS3D descriptors used in GA-PLS models include information corresponding to hydrogen-bond acceptor interactions (CATS3D_08_AA) and lipophilic character of a molecule (CATS3D_03_LL).

## 3. Materials and Methods

### 3.1. Reagents

Methanol HPLC grade (99.9%), sodium dodecyl sulphate (SDS), 4-(2-Hydroxyethyl)piperazine-1-ethanesulfonic acid (HEPES), tris(hydroxymethyl)aminomethane (TRIS), and thiourea were purchased from Sigma-Aldrich (Steinheim, Germany). Dimethyl sulfoxide (DMSO) and sodium hydroxide (NaOH) were bought from POCH (Gliwice, Poland). Ultrapure water was purified by Millipore Direct-Q 3 UV Water Purification System (Millipore Corporation, Bedford, MA, USA).

### 3.2. Analytes

The synthesis of the investigated compounds is previously reported [15,21,22]. Briefly, compound **14** was obtained via condensation of *N*-hydroxy-3-(hydroxyamino)-3-iminopropanamide with acetylacetone in the presence of piperidne [15], while compounds **1** and **23** were synthesized through multi-step procedure from aniline and 2,5-dimethoxyaniline, respectively [21,36]. Derivatives **2–5** and **6–13**, **15–18** were synthesized by acylation, alkylation or sulfonation reactions of compounds **1** [15,22] and **14** [21], respectively, while compounds **19–22**, **24–26** were obtained through methylation of the corresponding isoxazolones [21]. Chemical names and the structures of the investigated compounds are presented in Appendix A whereas SMILES notation is listed in supplementary data excel sheet file (in the worksheet SMILES notation). The target compounds were dissolved in DMSO to obtain a concentration of 1 mg mL^−1^. The stock solutions of analytes were stored at 2–8 °C prior to analyses.

### 3.3. RP-TLC Analysis

RP-TLC experiments were performed with ready to use C_18_ and C_8_ plates (20 cm × 10 cm) manufactured by Merck (Darmstadt, Germany) with F_254_ fluorescence indicator. The chromatographic chambers (Twin Trough Chambers from CAMAG, Philadelphia, PA, USA) were saturated with the mobile phase vapors for 30 min. The 5 µL of the stock solutions of the analytes were spotted manually on the plates with the use of a micropipette from Brand (Wertheim, Germany). The mobile phases were prepared by mixing appropriate volumes of methanol and water in a range from 40 to 90% (*v*/*v*), in the case of C_8_ bonded silica, and from 60 to 100% (*v*/*v*), in the case of C_18_ bonded silica stationary phase. In each chromatographic experiment, the content of organic modifier was increased in steps of 10% (*v*/*v*). Chromatograms were developed at a room temperature (20 ± 2 °C) in ascending fashion to the solvent distance of 8 cm. Then, the chromatographic plates were dried in a stream of warm air for 5 min. The identification was performed under UV light at λ = 254 nm by CAMAG UV Lamp 4 and the Viewing Box 4 (Philadelphia, PA, USA).

Subsequently, the Soczewiński–Wachtmeister’s [37] equation, which presents linear relationship between the concentration of organic solvent in mobile phase (*C*) and retention factor *R*_M_,
(4)$$R_{M}=R_{M}{}^{0}-mC,$$
was used in order to determine the basic lipophilicity RP-TLC parameter *R*_M_^0^ and hydrophobic constant *m*. Another parameter of RP-TLC lipophilicity, *C*_0_, introduced by Bieganowska [38], has been calculated according to the formula

(5)$$C_{0\;}=\;\frac{R_{M}^{0}}{m}$$

This metric corresponds to the parameter φ_0_ (the isocratic chromatographic lipophilicity index) previously intended for the HPLC technique. *C*_0_ parameter relates to the concentration of the organic component in the mobile phase for which the distribution of the analyzed substance between the mobile and stationary phase is equal (1:1) [38].

Moreover, the PCA was performed according to the protocol proposed by Sarbu and co-workers [39] to calculate the principal component PC_1_ as another estimate of TLC lipophilicity. The data matrixes included *R*_M_ values of solutes × modifier concentrations in the mobile phase.

### 3.4. HPLC Analysis

During this study, we used a Prominence-1 LC-2030C 3D HPLC system (Shimadzu, Japan) equipped with DAD detector and controlled by LabSolution system (version 5.90 Shimadzu, Japan). The concentrations of the investigated analytes were approximately 100 μg/mL, and the injected volume was 20 μL. The RP-HPLC experiments were performed on Knauer C_18_ 100 × 4.6 × 5 µ HPLC column with a linear gradient 20–98% phase B (where phase A was water and phase B was methanol) at a flow rate of 1 mL/min. The temperature of the chromatographic column was controlled and set to 30.0 °C. Two gradient runs differing in gradient time (*t*_G_ equal to 20 min and 40 min) were performed and retention times (*t*_R_) of investigated isoxazolone derivatives were collected. Detection of solutes was performed at 290 nm. These data were used as input data, and appropriate log*k*_w_ values (i.e., the retention factor log*k* extrapolated to 0% organic modifier, as an alternative to log*P*) were calculated using the DryLab 6.0 software (Molnar Institute, Berlin, Germany) based on the assumption proposed by Snyder and co-workers [40,41]. Each HPLC run was repeated twice. Dwell volume for HPLC system was measured at 0.780 mL, whereas the obtained dead time for used HPLC columns was equal 1.401 min.

### 3.5. MEKC Analysis

All MEKC experiments were carried out with a P/ACE MDQ plus system (Sciex, Framingham, MA, USA). The electropherograms were recorded and analyzed with the 32 Karat Software (version 10.2). The uncoated fused silica capillaries (50 mm i.d., Polymicro Technologies, West Yorkshire, UK) of a total length equal to 60 cm × 50 µm were used during the study. The following rinsing procedures were applied before every working day: first rinsing with 0.1 M NaOH for 30 min, next ultrapure water for 10 min, and finally BGE for 30 min. Between analyses the capillary was conditioned with BGE for 2 min. The applied pressure for all rinsing operations was 345 kPa. The investigated isoxazolone derivatives were dissolved in BGE at concentrations of 100 µg/mL with the addition of quinine (micelles marker) and DMSO (EOF marker). Hydrodynamic injected mode (35 kPa for 5 s) was used to introduce samples into the capillary. The separation condition was as follows: voltage application of 20 kV with positive polarity and a constant temperature of 25 ± 0.1 °C. The separations were performed in duplicates. The BGE consisted of aqueous solution of 50 mM SDS and 120 mM HEPES/100 mM Tris buffer of pH 7.4. Detection was carried out at 200, and 250 nm with 8 Hz probing frequency. The logarithm of retention factor log*k*_MEKC_ was calculated by the equation proposed by *Terabe* and co-workers.

(6)$$\log k_{MEKC}=\log\left(\frac{t_{R}-t_{EOF}}{t_{EOF}\left(1-t_{R}/t_{MC}\right)}\right)$$

### 3.6. Calculated Lipophilicity

Several software packages with different algorithms were used for lipophilicity calculations. Six different log*P* values (Alog*P*s, AClog*P*, Alog*P*, MLOGP, Xlog*P*2, Xlog*P*3) and average log*P* (Avg log*P*) were obtained with Virtual Computational Chemistry Laboratory (VCCLAB, http://www.vcclab.org/) accessed on 21 September 2019. The miLog*P* values were established using the Molinspiration algorithm (http://www.molinspiration.com/), while KOWWIN log*P* values were computational using KOWWIN v. 1.68 software (EPI Suite package v.4.1, U.S. EPA). Lipophilicity parameters log*P*C and log*P*V (am1) implemented into the Spartan’08 package (www.wavefun.com) were calculated according to the models of Ghose, Pritchett, and Crippen [42] and Villar [43,44], respectively. Additional log*P* and Clog*P* indexes were derived from ChemDraw suite. Vlog*P* parameter was obtained using Bernard Testa’s virtual log*P* calculator available online: https://nova.disfarm.unimi.it/vlogp.htm (accessed on 21 September 2019). All the calculated log*P* values are summarized in Table 1.

### 3.7. DFT Calculations

All calculations have been performed with the Gaussian 16 package using default thresholds and algorithms. The standard hybrid Becke-3–Lee-Yang-Parr functional (B3LYP) [45], parameter-free Perdew–Burke–Enzerhof (abbreviated as PBE0 or PBE1PBE) functional [46], long-range-corrected hybrid functionals CAM-B3LYP [47] and ωB97XD [48] were utilized. The bulk solvent effects were taken into account for the DFT calculations by means of Solvation Model based on Density (SMD) [49]. A selection of standard basis sets has been used in the course of this study including the 6-31G+, 6-311G+(d,p), and 6-311G++(2df, 2dp). The geometry optimizations of all molecules in their ground states were carried out with the inclusion of solvent effects. Vibrational analysis was used to verify that the optimized structures correspond to local minima on the energy surface. Gibbs free energies, including zero-point corrections, temperature corrections, and vibrational energies, were computed for standard conditions (T = 298.15 K, P = 1.0 atm) using the harmonic oscillator approximation. The theoretical logarithm of partition coefficient (log*P_DFT_*) was calculated according to the following equation:(7)$$\log P_{DFT}=\left(\frac{\Delta G_{water}-\Delta G_{n-octanol}}{2.303RT}\right)\;.$$

The calculated log*P_DFT_* values are presented in Table 3.

### 3.8. Principal Component Analysis (PCA)

Principal component analysis (PCA) was performed for two databases using the Statistica 10 software package (StatSoft, Tulsa, OK, USA). The first set of data included the obtained data matrixes (solutes × modifier concentrations) from TLC separations. The seconded investigated PCA matrix included lipophilicity indices obtained with a variety of indirect methods, theoretical methods, and antifungal activities. Due to the fact that these data have different units, they were standardized to unit variance and zero mean in order to eliminate the impact of divergent scales.

### 3.9. Sum of Ranking Differences (SRD) Analysis

Sum of ranking differences (SRD) was performed on lipophilicity data using Microsoft Excel visual basic macros freely available from http://aki.ttk.mta.hu/srd/. Although the SRD methodology is described in details by Hébrger and Hunek in their source papers [50,51,52], here, we will provided a short summary. During the SRD analysis, the studied compounds and lipophilicity estimation methods are arranged in rows and columns of a data matrix, respectfully. Since different lipophilicity measures were expressed in different scales, before the SRD analysis, they were transformed to the same scale. Since there are many methods to rescale the data, we have decided to test three most frequently employed strategies: standardization, i.e., scaling to the unit standard deviation (STD), interval scaling between 0 and 1 (IS), and rank transformation (RNK). In order to rank and compare different lipophilicity methods, SRD requires an additional column in a data matrix, the so-called reference vector. This vector can be a series of row-wise minima, maxima, or arithmetic means pulled from the all studied methods, or it can be a series of gold standard reference values. In this particular case, we have used a series of arithmetic means as a benchmark (consensus-based comparison). Consensus-based comparison has several advantages. First, every lipophilicity estimation method suffers from systematic and random errors. These errors are at least partially cancelled out by calculating the arithmetic mean. Second, the arithmetic mean, according to the maximum likelihood principle, is the value accompanied by maximum probability to be considered as a true value. After adding a proper reference column, all the values in the data matrix are ranked column-wise, and the ranks associated with each method are subtracted from the reference ranks. The obtained differences are then summed up in SRD values, which are then normalized according to Equation (8) and associated with each method. The smaller the SRD value is, the better is the lipophilicity estimation method, i.e., the closer is to the reference.

(8)$$RD\left(\%\right)=\frac{SRD}{SRD_{max}}\times 100$$

In order to validate such ranking, two approaches are implemented. One is based on comparison with a random distribution of SRD values, comparison with random numbers (SRD-CRRN), which tests the null hypothesis that ranking of objects by each particular method is performed by a chance. If the SRD value of a method lies within the random distribution curve, then the null hypothesis cannot be rejected and the ranking of compounds by that particular lipophilciity method is statistically insignificant at the 95% confidence level. Otherwise, methods lying out of the random distribution curve are statistically significant. The other approach is a sevenfold jack knife–like cross-validation procedure. It removes 1/7 of objects out of a data matrix and performs the SRD analysis. The procedure is repeated seven times, resulting in seven SRD values associated with each method. Methods are then arranged in ascending order of SRD medians and depicted in the form of a box plot, and similar ones are grouped into sections following non-parametric pairwise comparison testing. STATISTICA 9.1 (Statsoft, Tulsa, OK, USA) was further used for non-parametric significance testing, as well as the analysis of variance (ANOVA) of the SRD values obtained by the sevenfold cross-validation.

### 3.10. QSRR Analysis

In order to get insight into molecular mechanisms governing retentions and predict the four end points in the studied chromatographic systems (*R_M_*^0^ for C_8_ and C_18_ TLC systems, log*k*_w_ for HPLC, and log*k* for the MEKC system) QSRR analysis was performed. Briefly, the QSRR approach was proposed by *Kaliszan* at the end of the 1970s [3,53]. This approach analyzes the influence of the molecular structure of the analytes on their respective retention factors.

Molecular descriptors were calculated for structures optimized in water by means of DFT at the PBE0/6-311G++(2df,2pd) level [46] and the SMD solvation model [49] (files available in mol format) using the Dragon 7.0 (Talete, Milan, Italy) software. Full names, symbols and definitions of the descriptors can be found the handbook by Todeschini et al. [54]. In total, 2848 descriptors were calculated for 26 molecular structures.

For construction of the QSRR models, the initial matrix of molecular descriptors was notably reduced using stringent pre-selection criteria, removing all the molecular descriptors with (i) missing values, (ii) relative standard deviation value < 5%, and (iii) those that were found to be redundant (pairwise *R* > 0.5 and <−0.5, removed the descriptor less correlated with the dependent variable) [55]. The dataset was split into a training and an external validation set (70/30% proportion) using the Kennard and Stone algorithm [56]. Genetic algorithm in its binary formulation coupled with partial least squares (GA-PLS) [55,57,58] was used for simultaneous variable selection and retention factor modeling. Each unit of a GA population comprised of molecular descriptors encoded in binary format (1: selected, 0: not selected). GA hyper-parameters and functions were set as follows: population size of 20, cross-over fraction of 0.8 with using the single-point function, mutation rate of 0.2 using the uniform mutation function, and tournament selection function. Leave-one-out cross-validation was used to optimize the number of latent variables within each unit and for the final models. The objective function of the GA was root mean square error of cross-validation,
(9)$$RMSECV=\sqrt{\frac{\sum_{i=1}^{n}{\left(y\left(cv\right)-y\left(\exp\right)\right)}^{2}}{n}},$$
where *y*(exp), *y*(cv) represent the experimental values of the dependent variable and those estimated using LOO-CV, respectively. The GA-PLS algorithm was performed in 1000 cycles for each chromatographic setup. Final consensus GA-PLS models were built out of the molecular descriptors with the percentage of selection (% selection) higher than the mean % selection of all the molecular descriptors. Cross-validated analysis of variance (CV-ANOVA) [59] was employed to test the statistical significance of all the final consensus GA-PLS models. All the calculations pertaining to the QSRR analysis were performed in MATLAB 2019b (Mathworks, Sherborn, MA, USA).

## 4. Conclusions

Lipophilicity is an essential parameter for selection of compounds which may constitute a starting point for the development of novel antifungal drug candidates. Drug discovery aims to achieve strong potency with a minimal increase in molecular weight or lipophilicity. Among the studied compounds, structure **16** can be considered as the best starting structures for further studies, since it demonstrates the lowest lipophilic character among the compounds with pronounced antifungal activity.

The comparison of methods applied for lipophilicity assessment proved that three computational indices, i.e., Vlog*P*, AClog*P*, log*P* Chem Draw, along with MEKC are the least applicable methods for lipophilicity determination of the investigated pyrido-isoxazolones. According to the obtained SRD analysis, it is not possible to give a clear recommendation for any particular computational program, but the lowest SRD scores indicated that the chromatographic lipophilicity parameters obtained using C_8_ and C_18_ modified silica, regardless to the chromatographic technique employed (TLC or HPLC), should be considered as the best manner of lipophilicity estimation for the studied pyrido-isoxazolones.

## Figures and Tables

**Figure 1 molecules-24-04311-f001:**
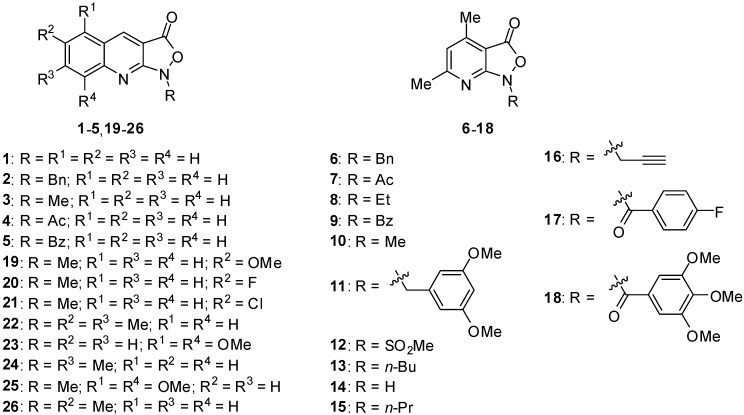
The investigated pyrido- and quinolino-isoxazolones.

**Figure 2 molecules-24-04311-f002:**
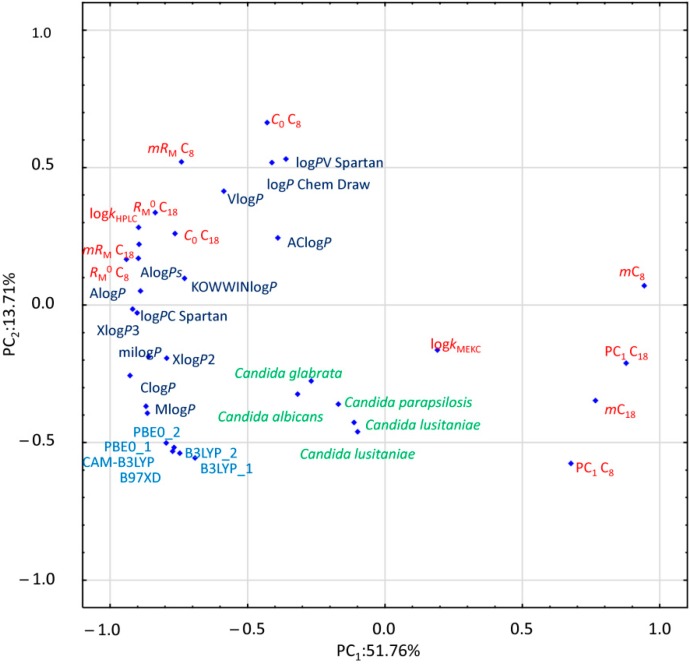
Principal component analysis (PCA) loading plot. The chromatographic data are marked in red, the calculated log*P*s are presented in dark blue, DFT parameters are marked in light blue, and the biological data are signed in green. DFT labels include B3LYP_1: B3LYP/631G+, B3LYP_2: B3LYP/6311G++dp, CAM-B3LYP: CAM-B3LYP/6311G++dp, B97XD: ωB97XD/6311G++dp, PBE0_1: PBE0/6311G++dp, and PBE0_2: PBE0/6311G++2df2pd.

**Figure 3 molecules-24-04311-f003:**
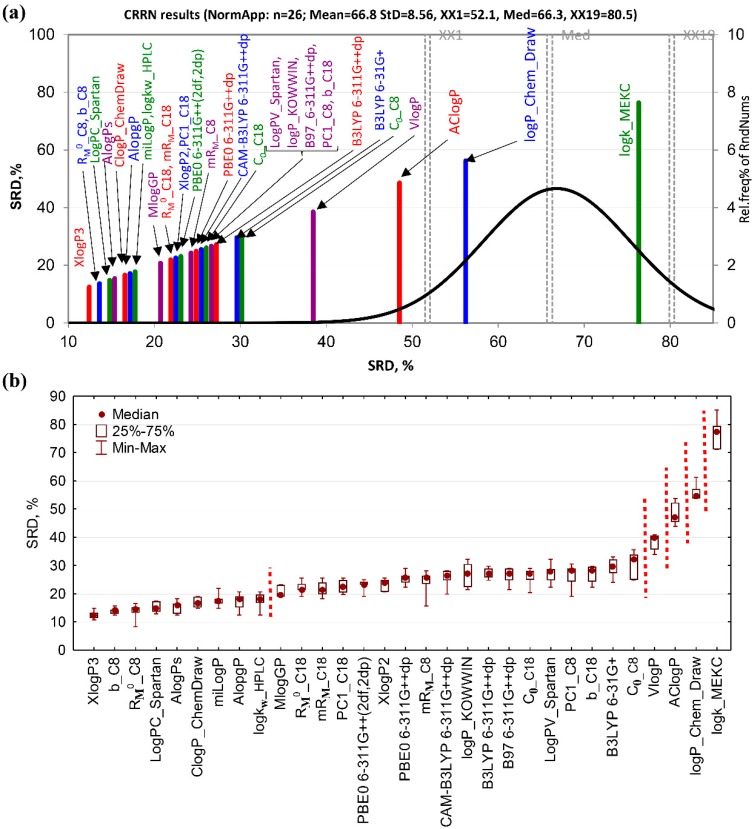
Consensus-based ranking of chromatographic and computational lipophilicity measures. (**a**) SRD-CRRN of standardized lipophilicity measures; the SRD values are depicted on *x* and *y*-axis, *n* = 26, random distribution Mean = 66.8, *StD* = 8.56, *XX*1 = 52.1, Med = 66.3, *XX*19 = 80.5, (**b**) box and whisker plot of normalized SRD values obtained by the sevenfold cross-validation. Indices for which the median SRD values are statistically significantly different at the predefined significance level of *p* = 0.05 (tested by both, the sign test and the Wilcoxon’s matched pairs test) are separated by dashed lines.

**Figure 4 molecules-24-04311-f004:**
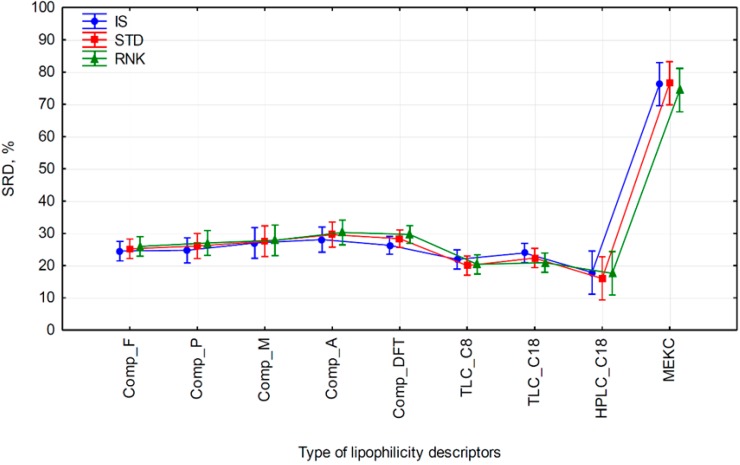
Factor effects presented as level arithmetic means and 95% confidence intervals (denoted as vertical bars). SRD score values are plotted on *y*-axis. *F*_1_ is depicted as lines of different colors, while *F*_2_ is plotted on *x*-axis.

**Figure 5 molecules-24-04311-f005:**
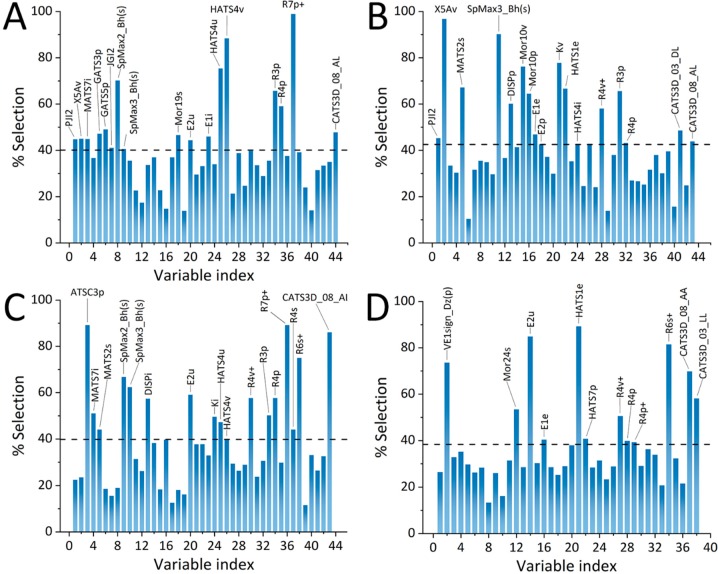
Percentage of selection (%Selection) of molecular descriptors for the final consensus GA-PLS QSRR models for (**A**) TLC C_8_, (**B**) TLC C_18_, (**C**) HPLC log*k*_w_, and (**D**) MEKC log*k* parameters.

**Figure 6 molecules-24-04311-f006:**
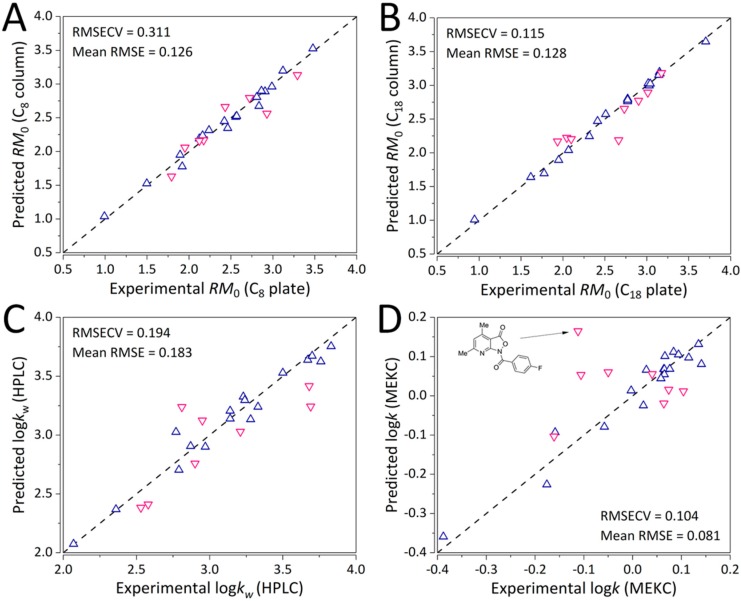
Predictive ability of the final consensus GA-PLS QSRR models for (**A**) TLC with the C_8_ plate, (**B**) TLC with the C_18_ plate, (**C**) HPLC log*k*_w_, and (**D**) MEKC log*k* parameters. Upwards-pointing triangles (blue) refer to the training set, whereas the downwards-pointing triangles (pink) refer to the external validation set analytes.

**Table 1 molecules-24-04311-t001:** The calculated log*P* values of the investigated compounds with respect to the computational model.

No	log*P* Chem Draw	Clog*P* Chem Draw	vlog*P*	log*P*C Spartan	log*P*V Spartan	miLog*P*	log*P* KOWWIN	Alog*P*s	AClog*P*	Alopg*P*	Mlog*P*	Xlog*P*2	Xlog*P*3
1	2.36	1.00	2.13	1.94	0.03	1.50	1.69	1.26	3.45	2.02	2.07	2.33	2.33
2	2.82	3.83	2.96	4.47	2.41	3.16	3.94	3.67	3.37	3.95	3.70	3.44	3.96
3	2.27	2.06	2.31	2.73	0.46	1.57	2.23	2.08	2.01	2.37	2.37	1.73	2.47
4	0.32	1.76	1.03	1.59	0.82	1.82	0.28	1.33	2.04	1.53	2.18	1.72	1.70
5	1.61	4.10	2.02	3.49	1.43	3.04	1.88	2.71	3.52	3.20	3.51	3.44	3.36
6	2.09	3.44	2.99	3.73	1.64	2.76	3.70	3.20	2.72	3.38	3.36	2.84	3.39
7	−0.42	1.37	0.95	0.86	−0.71	1.42	0.03	0.72	1.39	0.96	1.77	1.12	1.13
8	1.07	2.20	2.01	2.34	0.15	1.54	2.48	2.14	1.78	2.15	2.24	1.55	2.26
9	0.87	3.72	1.93	2.76	0.81	2.63	1.64	2.11	2.86	2.63	3.17	2.84	2.79
10	1.53	1.67	2.32	2.00	−0.20	1.16	1.99	1.55	1.35	1.80	1.94	1.13	1.89
11	2.35	3.45	3.33	3.48	1.12	2.80	3.86	3.25	2.51	3.35	2.83	2.67	3.33
12	1.19	0.11	1.87	0.69	4.72	0.54	−0.04	1.17	2.78	1.38	1.02	0.72	0.92
13	1.98	3.26	3.08	3.24	0.81	2.60	3.46	3.11	2.71	3.13	2.82	2.48	3.14
14	1.62	0.61	2.27	1.21	−0.60	1.09	1.44	1.27	2.79	1.45	1.62	1.72	1.76
15	1.52	2.73	2.59	2.82	0.52	2.04	2.97	2.58	2.25	2.67	2.53	1.91	2.79
16	0.91	1.77	1.83	2.22	0.64	1.32	2.20	1.59	1.24	2.99	2.45	1.40	2.01
17	1.01	3.86	2.26	2.91	0.87	2.80	1.84	2.43	2.92	2.83	3.56	3.00	2.89
18	2.48	2.74	3.43	3.35	0.72	2.39	3.21	3.22	3.40	3.34	2.57	2.18	3.30
19	2.40	2.35	2.55	2.61	0.21	1.60	2.31	2.08	1.90	2.35	2.13	1.65	2.44
20	2.41	2.27	2.60	2.89	0.66	0.73	2.43	2.41	2.06	2.57	2.78	1.89	2.57
21	2.89	2.84	3.01	3.29	1.36	2.22	2.88	2.62	2.62	3.03	2.91	2.35	3.10
22	3.10	3.01	3.31	3.71	1.06	2.37	3.33	2.74	2.63	3.34	2.92	2.61	3.20
23	2.62	1.58	2.67	1.69	−0.50	1.82	1.85	2.26	3.24	1.98	1.62	2.16	2.27
24	2.69	2.56	2.81	3.22	0.73	1.99	2.78	2.48	2.32	2.86	2.65	2.17	2.83
25	2.53	2.65	2.52	2.48	−0.12	1.89	2.40	2.34	1.80	2.34	1.89	1.56	2.41
26	2.69	2.56	2.99	3.22	0.79	1.99	2.78	2.48	2.32	2.86	2.65	2.17	2.83

**Table 2 molecules-24-04311-t002:** List of software used with information regarding algorithms and suppliers.

No.	log*P* Scale	Algorithms	Supplier
1	AClog*P*	atom-based method	http://www.acdlabs.com
2	Xlog*P*2	atom-based method	http://www.sioc-ccbg.ac.cn/
3	Xlog*P*3	atom-based method	http://www.sioc-ccbg.ac.cn/
4	Clog*P*	fragment contribution	http://www.biobyte.com (ChemDraw)
5	milog*P*	fragment contribution	http://www.molinspiration.com
6	Alopg*P*	fragment contribution	http://www.vcclab.org
7	log*P*C Spartan	fragment contribution	https://www.wavefun.com/
8	log*P*V Spartan	fragment contribution	https://www.wavefun.com/
9	log*P* Chem Draw	fragment contribution	http://www.perkinelmer.com
10	Alog*P*s	properties dependent methods (topological descriptors)	http://www.vcclab.org
11	Mlog*P*	properties dependent methods (topological descriptors)	http://www.talete.mi.it
12	KOWWINlog*P*	hybrid algorithm (atom-based approach and fragmental contribution)	http://www.epa.gov
13	Vlog*P*	properties dependent methods	https://nova.disfarm.unimi.it

**Table 3 molecules-24-04311-t003:** The calculated log*P* values of the investigated compounds with respect to the density functional theory (DFT) model.

No.	B3LYP 6-31G+	B3LYP 6-311G++dp	CAM-B3LYP 6-311G++dp	ωB97 6-311G++dp	PBE0 6-311G++dp	PBE0 6-311G++(2df,2dp)
1	−1.47	−0.79	−0.78	−0.74	−0.70	−0.54
2	1.48	1.82	1.68	1.81	1.93	2.06
3	0.02	0.36	0.30	0.30	0.45	0.59
4	−0.43	0.24	0.00	0.35	−0.50	0.68
5	0.71	1.48	1.58	1.80	1.55	1.58
6	1.69	1.85	2.16	1.77	1.93	2.08
7	0.22	1.35	1.65	1.54	1.11	−0.06
8	0.57	0.85	0.77	0.79	0.94	1.10
9	0.80	1.40	1.21	1.26	1.57	1.62
10	−0.09	−0.24	−0.31	−0.28	−0.16	−0.01
11	−0.70	0.90	0.70	0.85	0.05	0.25
12	−2.58	−1.70	−1.81	−1.76	−1.57	−1.26
13	1.55	2.00	2.05	1.88	1.99	2.08
14	−1.94	−1.29	−1.31	−1.26	−1.23	−1.10
15	0.64	1.47	1.36	1.42	1.54	1.68
16	−0.18	0.15	0.01	0.07	0.22	0.38
17	0.64	1.37	1.33	1.47	1.43	1.58
18	−1.29	0.01	0.57	0.55	0.48	1.14
19	−0.76	−0.07	−0.11	−0.10	0.07	0.30
20	0.01	0.39	0.32	0.30	0.52	0.67
21	0.50	0.77	0.76	0.78	0.89	1.04
22	0.71	1.03	0.98	0.93	1.11	1.25
23	−1.86	−1.86	−0.68	−0.62	−0.56	−0.32
24	0.43	0.65	0.63	0.63	0.79	0.96
25	−1.16	−0.32	−0.30	−0.29	−0.13	0.28
26	0.34	0.70	0.64	0.64	0.79	0.94

**Table 4 molecules-24-04311-t004:** Chromatographically determined lipophilicity indices.

No.	TLC: C_8_ Bonded Silica Gel					TLC: C_18_ Bonded Silica Gel					HPLC	MECK
	*R* _M_ ^0^	*m*	*C* _0_	m*R*_M_	*PC*_1_ *	*R* _M_ ^0^	*m*	*C* _0_	m*R*_M_	*PC*_1_ **	log*k*_w_	log*k*
1	1.496	2.198	0.681	0.068	3.593	2.771	3.153	0.879	−0.266	3.952	2.360	0.135
2	3.120	3.730	0.836	0.696	−2.265	3.702	4.031	0.918	0.477	−2.999	3.980	−0.058
3	2.162	2.805	0.771	0.339	1.033	2.045	2.414	0.847	0.114	0.130	2.870	−0.106
4	1.893	2.593	0.730	0.208	2.319	1.935	2.440	0.793	−0.017	1.508	2.790	0.141
5	2.929	3.592	0.815	0.594	−1.271	3.143	3.561	0.882	0.294	−1.278	3.760	−0.159
6	3.293	4.063	0.810	0.652	−1.648	3.012	3.363	0.896	0.322	−1.656	3.680	−0.050
7	1.791	2.477	0.723	0.182	2.530	1.616	2.087	0.774	−0.054	1.742	2.530	0.066
8	2.175	2.772	0.785	0.374	0.641	2.098	2.465	0.851	0.126	0.022	2.900	0.074
9	2.834	3.539	0.801	0.533	−0.648	2.901	3.345	0.867	0.225	−0.648	2.770	0.058
10	1.919	2.558	0.750	0.257	1.779	1.775	2.186	0.812	0.026	0.952	2.580	0.040
11	3.477	4.192	0.829	0.752	−2.635	3.041	3.324	0.915	0.382	−2.284	3.830	0.094
12	2.238	2.094	1.068	0.876	−5.068	3.017	3.633	0.830	0.111	0.563	3.140	0.064
13	2.988	3.555	0.840	0.677	−2.168	3.175	3.457	0.918	0.409	−2.514	3.690	−0.003
14	0.991	1.831	0.541	−0.199	6.251	0.945	1.841	0.513	−0.528	6.519	2.070	0.065
15	2.722	3.391	0.803	0.518	−0.571	2.770	3.111	0.890	0.281	−1.322	3.280	0.104
16	1.951	2.734	0.714	0.174	2.758	1.948	2.641	0.737	−0.165	3.120	2.810	0.084
17	2.864	3.579	0.800	0.538	−0.681	3.158	3.649	0.865	0.238	−0.656	3.700	−0.112
18	2.808	3.476	0.808	0.548	−0.832	3.151	3.743	0.842	0.157	0.226	3.670	0.022
19	2.421	3.023	0.801	0.456	−0.103	2.509	2.888	0.869	0.198	−0.561	3.140	−0.388
20	2.123	2.780	0.764	0.317	1.259	2.314	2.778	0.833	0.092	0.493	2.970	0.076
21	2.568	3.182	0.807	0.500	−0.491	2.412	2.718	0.887	0.238	−1.021	3.330	0.065
22	2.907	3.498	0.831	0.634	−1.748	3.028	3.319	0.912	0.373	−2.206	3.500	0.115
23	2.123	2.664	0.797	0.391	0.403	2.066	2.569	0.804	0.011	1.248	2.950	−0.176
24	2.431	3.033	0.801	0.459	−0.127	2.731	3.107	0.879	0.245	−0.966	3.210	0.028
25	2.557	3.026	0.845	0.591	−1.536	2.664	2.980	0.894	0.280	−1.377	3.230	−0.161
26	2.461	2.990	0.823	0.518	−0.773	2.771	3.153	0.879	0.249	−0.987	3.240	0.064

*R*_M_^0^—extrapolated value of *R*_M_ to 0% organic modifier, *m*—hydrophobic constant; *C*_0_—RP-TLC lipophilicity indexes introduced by Bieganowska; *m*R_M_—mean value of *R*_M_; *PC*_1_—first principal component of data matrixes included *R*_M_ values of solutes × modifier concentrations in mobile phase; log*k*_w_—retention factor log*k* extrapolated to 0% organic modifier determined by HPLC; log*k*_MEKC_ retention factor determined by MEKC. * contains 88.489% of information of retention matrix; ** contains 91.019% of information of retention matrix.

**Table 5 molecules-24-04311-t005:** Statistical significance of factor effects based on ANOVA of 651 SRD score values collected in the sevenfold cross-validation.

	Degrees of Freedom	*SS*	*MS*	*F*	*p*
**Intercept**	**1**	**394,895.4**	**394,895.4**	**4833.701**	**0.000000**
*F* _1_	2	11.7	5.8	0.071	0.931134
***F*_2_**	**8**	**59,022.7**	**7377.8**	**90.308**	**0.000000**
*F*_1_ × *F*_2_	16	616.3	38.5	0.471	0.960351
Error	624	50,978.5	81.7		
Total	650	110,654.1			

Factors: *F*_1_—type of data pretreatment: standardization (STD), interval scaling (IS), rank transformation (RNK); *F*_2_—types of lipophilicity descriptors: computational methods based on atomic (Comp_A) and fragment contributions (Comp_F), property-based (Comp_P), hybrid methods (Comp_M), DFT based calculations (Comp_DFT), chromatographic indices obtained from TLC experiments on C_8_- (TLC_C8), C_18_-modified silica (TLC_C18), calculated from HPLC experiments on C_18_ modified silica (HPLC_C18), as well obtained with micellar electrokinetic chromatography (MEKC) study. Significant factors are indicated in bold.

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
