# Peer review of "Lipophilicity Determination of Antifungal Isoxazolo[3,4-b]pyridin-3(1H)-ones and Their N1-Substituted Derivatives with Chromatographic and Computational Methods"

_molecules, 2019, doi:10.3390/molecules24234311_

Round 1

Reviewer 1 Report

The manuscript  by Ciura et al. deals with determination of lipophilicity of 26 compounds by various methods (experimental and computational). The manuscript is interesting but needs deeper discussion of obained results. Many methods were used and the results are presented but they are not related to the structure of the tested compounds.

Specific comments:

1) Abbreviations must be defined upon first appearance.

2) In the "Results and Discussion" part the wider comparison of experimentally and theoretically obtained values of lipophilicity should be added.

3) I miss some discussion about determined lipophilicity and structure of the tested compounds.

4) Lines 82-87 better fits to Introduction.

5) Tested functionals od DFT calculations should be explained, for example as a footnote to Table 3.

6)Table 4 - all parameters (RM0, m....) should be explained under table.

7) I recommend to add general equations and explain all parameters before you discuss the results (i.e. equations from lines 475 and 479).

8)MEKC x MECK

9) Lines 230-232: Deeper discussion should be added.

10) Detection wavelength is missing at HPLC analysis.

Author Response

Reviewer 1

The manuscript  by Ciura et al. deals with determination of lipophilicity of 26 compounds by various methods (experimental and computational). The manuscript is interesting but needs deeper discussion of obained results. Many methods were used and the results are presented but they are not related to the structure of the tested compounds.

We would like to thank the Reviewer for valuable advices that greatly improved the manuscript. We referred to them in the following way.

Specific comments:

1) Abbreviations must be defined upon first appearance.

Author reply: According to the Reviewer's comment, these errors have been corrected.

2) In the "Results and Discussion" part the wider comparison of experimentally and theoretically obtained values of lipophilicity should be added.

Author reply: Taking into account the Reviewer's comments we deepened the discussion section. New information have been added into section 2.2. We do not change structures of proposed manuscript, since the information obtained after PCA and SDR would be repeated.  

3) I miss some discussion about determined lipophilicity and structure of the tested compounds.

Author reply: According to Reviser’s suggestion addition information have been added into section 2.2

4) Lines 82-87 better fits to Introduction.

Author reply: According to Reviewer's suggestion these sentences have been transferred to Introduction.

5) Tested functionals od DFT calculations should be explained, for example as a footnote to Table 3.

Author reply: According to Reviewer's suggestion we added relevant information: Section 3.7.

6)Table 4 - all parameters (RM0, m....) should be explained under table.

Author reply: According to Reviewer's suggestion we added relevant information.

7) I recommend to add general equations and explain all parameters before you discuss the results (i.e. equations from lines 475 and 479).

Author reply: We agree with the Reviewer that it would be clearer for the readers to introduce general equations and explain all the parameters before discussing the results. However, according to the guide for authors the entire “Materials and Methods” sections should be introduced at the end of the manuscript before the “Conclusions” section.

8)MEKC x MECK

Author reply: These mistakes have been corrected

9) Lines 230-232: Deeper discussion should be added.

Author reply: According to Reviewer's suggestion the discussion has been deepened in the context of the chemical structure of compounds.

10) Detection wavelength is missing at HPLC analysis.

Author reply: The missing information have been added.

Reviewer 2 Report

The submitted manuscript presents an interesting work concerning the determination of lipophilicity of new drug candidates with the use of chromatographic and computational methods and in my opinion it is fully suitable for the publication in Molecules journal.  One minor correction should be made before the publication of the manuscript: “Table 4” in line 335 (page 12) should be corrected to “Table 5”. 

Author Response

Reviewer 2

The submitted manuscript presents an interesting work concerning the determination of lipophilicity of new drug candidates with the use of chromatographic and computational methods and in my opinion it is fully suitable for the publication in Molecules journal.  One minor correction should be made before the publication of the manuscript: “Table 4” in line 335 (page 12) should be corrected to “Table 5”. 

We would like to thank Reviewer for taking the time to read the manuscript and express his/her opinion. The indicated mistake has been corrected.

Reviewer 3 Report

Dear Authors,

After reading your paper (I have to be honest, but it took a lot of time) I think finally I got to the essence of your paper in the conclusion. You have developed a computational study to assess how other software predict Log P, how your time consuming ab-initio methodology predict Log P in comparison with two methodologies to measure the lipophilicity of isoxazolo[3,4-b]pyridin-3(1H)-ones compounds with one conclusion. There are no clear computational methods that can help to know the “real” lipophilicity of those molecules except to the real experiment. So, a trivial conclusion. Despite that, in my personal opinion your work seems to be interesting and need a chance to be redeveloped. Here I provide you, if of course agree with them, let say a couple of advices and suggestions.

Best regards

Before starting a general comment here.

You lack originality. You have utilized different words but the same structure of this previous publication DOI: 10.3390/ijms20215288

My suggestion is to keep already in mind this information and do differently and to be honest even better. Moreover, is better to talk about the experimental methods previous to the computational ones. So please arrange the manuscript first developing by the experimental compound measurements. Then go for the computational study assessment of the log P.

Page 1, line 27: Fix the typo reserved with reversed.

Page 2, line 47 to 49: Why lipophilicity is one essential factor which determine the biological activity of a drug candidate? Please clarify and justify the sentence with some reference.

Page 2, line 51: Lipophilicity assessment remains a challenging task? Are you sure? If yes, please justify with a reference and develop more. To this reviewer, with the adventure of new UHPLC chromatography is it possible to run high throughput log P measurements.

Page 2, line 52: Please give a better definition of lipophilicity. Your definition is trivial and is not at all the correct one.

Page 2, line 55 to 57: Please add the appropriate references to RPLC and MECK. The Kalizsan reference is inappropriate to this reviewer. For the RPLC we recommend these two publications https://www.ncbi.nlm.nih.gov/pubmed/18189348

https://link.springer.com/article/10.1365/s10337-005-0608-6

Page 2, line 58 to 59: Why you state that in-silico approaches dedicated for log P calculation are intensively developed? What is the need for this sentence? I think is a bit out of context.

Page 2, line 59 to 63: Why ab-initio calculations are useful compared to the aforementioned log P software? Why people should use that? Not clear this statement. Please rewrite and develop.

Page 2, line 66: Justify which “some of them showed a promising antifungal activity” with a reference. Moreover, be more specific. Which molecules shows antifungal activity? Who reported that?

Page 2, line 66 to 68: Please provide a reference for this sentence.

Page 2, line 70: Several methods?? Which ones? Please be specific and clear and add a reference.

Page 2, line 85 to 87: Please provide a reference for this sentence.

Page 3, line 93 to 97: Maybe this part belongs to the introduction in order to justify why computational methods are useful despite the high-throughput log P determination methods. Please rework the manuscript to be clear, especially in the introduction and to better develop the aim of this work.

Page 3, line 99 to 102: Concern in regard to Table 1 and Table S1 and Table S2. I do not see the advantage to keep the 2D molecular structure in the supplementary material. My suggestion is to make a table with all the 2D structures in this manuscript. Scientists need to see the molecular structures along with Log P determinations. Indeed, you add min and max value of Log P to show how different they are. The reviewer suggests adding the average and standard deviation and also to describe why these methods shows different lipophilicity results.

Page 3, line 106: Correlation matrix was not specified where is it. Is there, but please specify the table number, file in supplementary material and so on. Please fix this.

Page 4, line 113 to 114: Mean value in respect to what? Please develop.

Page 4, line 125: Spartan is also a theoretical method true? So your criticism is against all the aforementioned computational methods to calculate log P? This sentence is not clear to this reviewer.

Page 5, line 133 to 135: I do not understand. Previously you said that computational log P estimation are bad. Now you state that better computational method can be utilized for that, the ab-initio calculation. Why? Why I should use a such long and time-consuming computational method despite the QSPR previous methods? What is the need for that? Is the result significantly different from the other computational methods and nearer to the “ground true”?

Page 5, Table 3. Same story of table 1. Add the mole

Page 5, line 142 to 143: Where is log kw in table 3?? Please add this value.

Page 6, line 149 to 152: What do you mean with “self-learning” algorithms? What is sentence state for? Please remove. It sounds odd.

Page 6, line 156: This is not the right reference. Please correct this with the previous one that I gave to you (Carrupt et all).

Page 6, line 160: No reference here for MECK?

Page 6, line 161 to 162: Why HPLC and MECK are different? Which one is the correct measurement?

Page 6, Table4: Very poor description! Please develop! What is m/C0 and so on. Which chromatographic methods are utilized in table 4? Develop.

Page 7, line 180 to 181: Slope of what? Please develop. Not clear.

Page 7, line 194 to 195: Where is located the correlation matrix? Again, please be more specific to which file belongs the correlation matrix. Moreover, how is calculated this correlation matrix?

Page 7, line 201: Why is so surprisingly?

Page 8, line 224 to 226: Please report the MIC value here in a table and add the doi for the source reference.

Page 9, line 268 to 269: Please add the acronym SDR here.

Pate 9, line 275 to 279: Calculation methodology not clear. Please develop more in the material and methods and remove this part from here. This is results.

Page 13, line 364: Would be interesting to see in the manuscript the PLS model with all the molecular descriptors as baseline to demonstrate that the GA-PLS improved the prediction ability.

Page 14, line 374: Please provide plot of LV v.s. RMSECV in supplementary materials.

Page 14 line 379: How did you selected the training and validation set? Kennard stones. Please specify here.

Page 14 and 15: these tables looks to this reviewer useless. Please move to the supplementary material.

Page 16, Figure of poor quality and not clear. What is in blue and what in pink? Train and validation?

Page 16, line 404 to 407: Please move this table in supplementary materials. To this reviewer, the information reported in these tables are not informative.

Page 20, line 408 to 457: Any mass spectra or NMR to verify the compound structure? If is not the case how do you certify the purity of the analyzed compounds?

Page 23, line 585 to 589: Please, can you provide SMILES and optimized MOL files? Others can benefit from your work.

Page 23, line 594 to 595: How? 80/20, 70/30? Please be more specific. Can you provide instead a leave one out predicted v.s. experimental instead of optimizing the parameters via LOO and then test against an external validation set? To this reviewer, a single validation set can present over fitting against itself especially in a such low number dataset.

Page 24, line 622 to 623: This sentence sounds odd. Is there any possible computational program able to predict well the log P among all the methodologies?

Please compare the predictions with the experimental values trough a t-test to say which computational methodology is better among all.

Author Response

Dear Authors,

After reading your paper (I have to be honest, but it took a lot of time) I think finally I got to the essence of your paper in the conclusion. You have developed a computational study to assess how other software predict Log P, how your time consuming ab-initio methodology predict Log P in comparison with two methodologies to measure the lipophilicity of isoxazolo[3,4-b]pyridin-3(1H)-ones compounds with one conclusion. There are no clear computational methods that can help to know the “real” lipophilicity of those molecules except to the real experiment. So, a trivial conclusion. Despite that, in my personal opinion your work seems to be interesting and need a chance to be redeveloped. Here I provide you, if of course agree with them, let say a couple of advices and suggestions.

Best regards

We thank the Reviewer for valuable advices that greatly improved the manuscript. We referred to them in the following way.

Before starting a general comment here.

You lack originality. You have utilized different words but the same structure of this previous publication DOI: 10.3390/ijms20215288

Author reply: The presented results relate to completely different compounds in terms of physicochemical properties, chemical structure, as well as biological activity. Previously reported quaternary (fluoro)quinolones are permanently charged, which significantly affects their chromatographic properties and in consequence their lipophilicity parameters.

The presented manuscript concerns dissimilar structures – isoxazolones.

Same parts of these two reports may seem similar, because both relate to comparison of methods for lipophilicity determination, however we emphasize that the study pertain to chemically different group of compounds. In the case of FQs, MEKC proved to be a very efficient tool, which do not correspond to the studied group of compounds. What's more, in the previously paper, SDR allowed for an unequivocal recommendation of certain calculation programs.

 In this work, we achieved several other/additional goals, such as:

QSRR analysis, which give the possibility of getting insight into the molecular mechanism of retention and interpretation of molecular descriptors which affect chromatographic lipophilicity indexes. application of ab-into methods for lipophilicity assessments designation of starting structures for further research on new antifungal drugs. correlation study between lipophilicity and antifungal activity

My suggestion is to keep already in mind this information and do differently and to be honest even better. Moreover, is better to talk about the experimental methods previous to the computational ones. So please arrange the manuscript first developing by the experimental compound measurements. Then go for the computational study assessment of the log P.

Author reply: We would like thank Reviewer for this suggestion, however we prefer to start the discussion with calculation methods as lipophilicity can already be roughly assessed at the stage of compound design prior to the synthesis. Hence it is more logical to start the discussion with computational methods.

Page 1, line 27: Fix the typo reserved with reversed.

Author reply: This mistake has been corrected.

Page 2, line 47 to 49: Why lipophilicity is one essential factor which determine the biological activity of a drug candidate? Please clarify and justify the sentence with some reference.

Author reply: According to Reviewer’s remark the additional information have been added.

Page 2, line 51: Lipophilicity assessment remains a challenging task? Are you sure? If yes, please justify with a reference and develop more. To this reviewer, with the adventure of new UHPLC chromatography is it possible to run high throughput log P measurements.

Author reply: According to the Reviewer’s remark this fragment has been reformulated.

Page 2, line 52: Please give a better definition of lipophilicity. Your definition is trivial and is not at all the correct one.

Author reply: According to the Reviewer’s recommendation the definition of lipophilicity has been corrected and definition presented in “Gold book” edited by UPAC has been quoted.   

Page 2, line 55 to 57: Please add the appropriate references to RPLC and MECK. The Kalizsan reference is inappropriate to this reviewer. For the RPLC we recommend these two publications https://www.ncbi.nlm.nih.gov/pubmed/18189348 https://link.springer.com/article/10.1365/s10337-005-0608-6

Author reply: According to the Reviewer’s recommendation the references have been changed.

Page 2, line 58 to 59: Why you state that in-silico approaches dedicated for log P calculation are intensively developed? What is the need for this sentence? I think is a bit out of context.

Author reply: We added this sentence to emphasize that this field is still being developed.

Page 2, line 59 to 63: Why ab-initio calculations are useful compared to the aforementioned log P software? Why people should use that? Not clear this statement. Please rewrite and develop.

Author reply: Ab initio calculations use wave functions to describe molecules. Since molecules and chemical bonds are in fact quantum phenomena (and not some arbitrary pieces of functional groups glued together with mathematical formula as in all other software) there is an obvious advantage of general DFT approach. Of course wave functions in DFT calculations only simulate Schrödinger equations, yet still they far better represent chemical species than the Euclidean models.

Page 2, line 66: Justify which “some of them showed a promising antifungal activity” with a reference. Moreover, be more specific. Which molecules shows antifungal activity? Who reported that?

Author reply: According to Reviewer’s comments appropriate references have been added. The information regarding molecules showing satisfactory antifungal activities are discussed on page 8 222-233. 

Page 2, line 66 to 68: Please provide a reference for this sentence.

Author reply: This part of introduction has been modified, the appropriate references have been added.

Page 2, line 70: Several methods?? Which ones? Please be specific and clear and add a reference.

Author reply: This part of introduction refer to applied method in presented study, all methods are listed in this part of introduction.

Page 2, line 85 to 87: Please provide a reference for this sentence.

Author reply: According to Reviewer’s suggestion the references have been added.

Page 3, line 93 to 97: Maybe this part belongs to the introduction in order to justify why computational methods are useful despite the high-throughput log P determination methods. Please rework the manuscript to be clear, especially in the introduction and to better develop the aim of this work.

Author reply: According to the Reviewer’s recommendation, this part have been transferred to introduction section.

Page 3, line 99 to 102: Concern in regard to Table 1 and Table S1 and Table S2. I do not see the advantage to keep the 2D molecular structure in the supplementary material. My suggestion is to make a table with all the 2D structures in this manuscript. Scientists need to see the molecular structures along with Log P determinations. Indeed, you add min and max value of Log P to show how different they are. The reviewer suggests adding the average and standard deviation and also to describe why these methods shows different lipophilicity results.

Author reply: The studied structures have been presented in Figure 1. The calculation methods do not provide values with standard deviation. We do not think that the average and standard deviations are relevant. Especially average value of any population may be extremely misleading. For example if a particular compound has logP values of 3.0, 2,9, 3,1 and 10 (according to various models) the average LogP is 4.75 (±3.5 ).

Page 3, line 106: Correlation matrix was not specified where is it. Is there, but please specify the table number, file in supplementary material and so on. Please fix this.

Author reply: We have added accurate information where the correlation matrix is available. Unfortunately, it is too big to be included to supplementary material, hence it is presented as separate data file.

Page 4, line 113 to 114: Mean value in respect to what? Please develop.

Author reply: According to the Reviewer’s suggestion this sentences has been corrected.  

Page 4, line 125: Spartan is also a theoretical method true? So your criticism is against all the aforementioned computational methods to calculate log P? This sentence is not clear to this reviewer.

Author reply: According to the Reviewer’s suggestion this sentences has been rewritten. 

Page 5, line 133 to 135: I do not understand. Previously you said that computational log P estimation are bad. Now you state that better computational method can be utilized for that, the ab-initio calculation. Why? Why I should use a such long and time-consuming computational method despite the QSPR previous methods? What is the need for that? Is the result significantly different from the other computational methods and nearer to the “ground true”?

Author reply: As we explained above ab initio approach is superior to the empirical approaches because it is more general. Currently DFT calculations are time consuming, however 20 years ago they were available for huge computational centers only. Nowadays these calculations are available for everyone (yes, they still take some time when using PC), however we believe that in the future computational time/power will not be an issue. Application of simple DFT approach proved to be as effective as use of empirical training-set-based models. We believe that having access to quick DFT calculations would allow for construction of LogP models that would be significantly more accurate than the current training-set-based models.

Page 5, Table 3. Same story of table 1. Add the mole

Author reply: The studied structures have been presented in Figure 1.

Page 5, line 142 to 143: Where is log kw in table 3?? Please add this value.

Author reply: Logkw is determined chromatographically – this information is included in this sentences and listed in table 4.

Page 6, line 149 to 152: What do you mean with “self-learning” algorithms? What is sentence state for? Please remove. It sounds odd.

Author reply: Self learning algorithms or artificial intelligence (AI) are based on artificial neural networks which automate the defining of logic by “learning” and adjusting weights of function approximations. In this case the algorithm would have to “learn” which DFT functional is best for logP prediction of any molecule featuring particular functional (chemical) groups.

Page 6, line 156: This is not the right reference. Please correct this with the previous one that I gave to you (Carrupt et all).

Author reply: The references has been changed.

Page 6, line 160: No reference here for MECK?

Author reply: The justification for using MECK was presented in the introduction with the appropriate citations.

Page 6, line 161 to 162: Why HPLC and MECK are different? Which one is the correct measurement?

Author reply: The issue is extensively discussed in the QSRR section, while information on which method can be recommended for the examined group of compounds is found in the conclusions.

Page 6, Table4: Very poor description! Please develop! What is m/C0 and so on. Which chromatographic methods are utilized in table 4? Develop.

Author reply:  The information have been added.

Page 7, line 180 to 181: Slope of what? Please develop. Not clear.

Author reply: Unfortunately the manuscript template for molecules indicates the experimental part is at the end of manuscript. In experimental part the SoczewiÅ„ski–Wachtmeister’s equation is presented.

Page 7, line 194 to 195: Where is located the correlation matrix? Again, please be more specific to which file belongs the correlation matrix. Moreover, how is calculated this correlation matrix?

Author reply: We have added accurate information where the correlation matrix is available. Unfortunately, it is too big to be included in supplementary material, hence it is presented in the data file. The correlation matrix has been calculated by Statistica software.

Page 7, line 201: Why is so surprisingly?

Author reply: Additional information has been added.

Page 8, line 224 to 226: Please report the MIC value here in a table and add the doi for the source reference.

Author reply: The appropriate reference are given, moreover the MIC values have been added to the supplement materials.

Page 9, line 268 to 269: Please add the acronym SDR here.

Author reply: The acronym SRD was added to the section title

Pate 9, line 275 to 279: Calculation methodology not clear. Please develop more in the material and methods and remove this part from here. This is results.

Author reply:  According to the Reviewer’s suggestion the entire section (from line 275 to 292) was removed and in changed form added to the experimental section. Please see the lines (579 – 582, 584, 587-597, 601-619). In this way, we hope that description of the SRD methodology was given in much clearer and easily comprehensible form. 

Page 13, line 364: Would be interesting to see in the manuscript the PLS model with all the molecular descriptors as baseline to demonstrate that the GA-PLS improved the prediction ability.

Author reply: After GA-PLS-based variable selection the predictive ability of the QSRR models may or may not improve with respect to the PLS model alone. The aim of variable selection was not solely to improve predictive ability, but to select the most informative descriptors and gain mechanistic insight into the RP-TLC and RP-HPLC separations of isoxazolo[3,4 b]pyridin 3(1H)-ones and their N1-substituted derivatives. Nevertheless, as a comparison the discussion was slightly extended with the results of the full PLS model. It can be located on page 13, line 380-393 in the revised version of the manuscript.

Page 14, line 374: Please provide plot of LV v.s. RMSECV in supplementary materials.

Author reply: Graphical depictions of the dependence between RMSECV and the number of LVs was added to SI (Figure 1S)

Page 14 line 379: How did you selected the training and validation set? Kennard stones. Please specify here.

Author reply: Analytes for the training and (external) validation sets were uniformly selected using the Kennard and Stone algorithm. In our opinion it is not appropriate to mention this here because it refers to the QSRR modeling methodology. Dataset separation was mentioned in section 3.10. (QSRR analysis) on page 18, line 262-263.

Page 14 and 15: these tables looks to this reviewer useless. Please move to the supplementary material.

Author reply: According to the Reviewer’s suggestion, these tables are transfered to SI.

Page 16, Figure of poor quality and not clear. What is in blue and what in pink? Train and validation?

Author reply: The “poor” quality is due to conversion to PDF format. It was improved in the revised version of the manuscript. The caption of this figure was also improved detailing which points correspond to the training, and which to the validation set.

Page 16, line 404 to 407: Please move this table in supplementary materials. To this reviewer, the information reported in these tables are not informative.

Author reply: According to the Reviewer suggestion, these tables are transfer to SI.

Page 20, line 408 to 457: Any mass spectra or NMR to verify the compound structure? If is not the case how do you certify the purity of the analyzed compounds?

The synthesis of the studied compounds has been published. See the section 3.2 

Page 23, line 585 to 589: Please, can you provide SMILES and optimized MOL files? Others can benefit from your work.

Author reply: According to the Reviewer’s suggestion we added SMILES and mol files in data file.

Page 23, line 594 to 595: How? 80/20, 70/30? Please be more specific. Can you provide instead a leave one out predicted v.s. experimental instead of optimizing the parameters via LOO and then test against an external validation set? To this reviewer, a single validation set can present over fitting against itself especially in a such low number dataset.

Author reply: The reviewer is correct. We have inadvertently omitted the proportion of

separation. The dataset was separated in a 70/30 % ratio and this is now noted in

parentheticals on page XXX line XXX of the revised manuscript. We respectfully disagree

that the parameters should not be optimized using LOO-CV. It was our intention to keep the

training and (external) validation sets entirely separate and thereby independent.

Performing LOO-CV on the entire dataset could lead to an underestimation or overestimation

of the model performance (in our work expressed as RMSECV). The reviewer is also correct

that validation of the QSRR models detailed in this manuscript would be more clear to the

readers if LOO-CV predictive ability plots were included. To prevent cluttering of the

manuscript, we have included them into the supplementary information as Figure 2S(A-D).

Upon doing so, there is a point that needed to be discussed. That is, the LOO-CV predictions

(on the training set) mostly agree with the ones presented in Figure 5. However, there is an

inconsistency for the MEKC QSRR model. LOO-CV has in this case potentially over-

estimated the training error, and the MEKC GA-PLS model should thereby be applied with

care. The QSRR analysis discussion was supplemented to reflect the above-mentioned, and is

located on page 17, lines 433-464.

Page 24, line 622 to 623: This sentence sounds odd. Is there any possible 

computational program able to predict well the log P among all the methodologies?

Author reply: According to the Reviewer’s suggestion, this sentences has been re-written. 

Please compare the predictions with the experimental values trough a t-test to say which computational methodology is better among all.

We would like thank the Reviewer for this suggestion, but we have decided to use extensive, robust, and sensitive non-parametric comparison of methods by the Sum of Ranking Differences which proved to be reliable in many occasions. Moreover, in this way we were able, in a single run, to rank all the methods, find the best and the worst ones, and group the similar ones into sections. Such grouping was done based on the sevenfold cross-validation followed by the two non-parametric methods for comparing statistically dependent samples (Sign test and Wilcoxon’s matched pairs test). Paired t-test, which would be only suitable test for this particular study, is less sensitive and have lower statistical power. Also, it assumes that values of differences among the reference and compared method are normally distributed. Such assumptions are frequently not satisfied. Also, in order to test all the methods, we should perform multiple comparison runs. Therefore, we would like to stay with current approach.

Round 2

Reviewer 1 Report

The manuscript has been corrected according to the comments. I have no additional comments and recommend manuscript for publication.

Reviewer 3 Report

Dear Authors,

you did an amazing job on the revised manuscript.

Best regards